# 5-Chlorocoumaranone-Conjugates as Chemiluminescent Protecting Groups (CLPG) and Precursors to Fluorescent Protecting Groups (FPG)



**Tim Lippold** [1], **Axel G. Griesbeck** [1,*] , **Robert Herzhoff** [2], **Mathias S. Wickleder** [3] , **Laura Straub** [3]
and **Niko T. Flosbach** [3]

1. Institute of Organic Chemistry, Department of Chemistry, University of Cologne, Greinstr. 4,
   50939 Köln-Cologne, Germany; lippoldt@smail.uni-koeln.de
2. Institute of Physical Chemistry, Department of Chemistry, University of Cologne, Greinstr. 4-6,
   50939 Köln-Cologne, Germany; rherzho1@smail.uni-koeln.de
3. Institute of Inorganic Chemistry, Department of Chemistry, University of Cologne, Greinstr. 6,
   50939 Köln-Cologne, Germany; n.flosbach@uni-koeln.de (N.T.F.); lstraub5@uni-koeln.de (L.S.);
   mathias.wickleder@uni-koeln.de (M.S.W.)
* Correspondence: griesbeck@uni-koeln.de

**Abstract:** The introduction and cleavage of protecting groups and the subsequent release of protected molecules is an important tool in synthetic organic chemistry. When polyfunctionalized substrates are involved, the reactivity of similar functional groups must be differentiated and selectively switched on and off. A very useful switching agent is visible or UV-light in photoremovable protecting groups (**PRPG**), allowing the PG release upon interaction with electromagnetic radiation. The reverse principle is the release of a protected molecule, which is accompanied by emission of light, i.e., chemiluminescent protecting groups (**CLPG**). This principle is proposed and investigated for phenylalanine (using ureido carboxylic acid **2** and its methyl ester derivative **3**) and the carbamate derivatives of paracetamol **4** and L-menthol **7**, protected as the corresponding urea-/carbamate-coumaranones **5A**, **5E**, **6** and **8**, respectively. While the carbamate derivative **6** released the protected substrate with a short and strong emission of blue light when treated with a base under atmospheric oxygen, **8** had to be treated additionally with potassium hydroxide in boiling ethanol to completely cleave the PG. Both urea-coumaranone derivatives **5A**/**5E** had a flash-like CL without release of the protected amino acid and, thus, were converted into a fluorescent protecting group (**FPG**).

**Keywords:** protecting group; amino acids; chemiluminescence; release

## 1. Introduction

Protecting and releasing group chemistry is an exceedingly important tool in synthetic organic chemistry [1–3]. Reactive groups are hampered in their reactivity by protecting groups and covered during specific reaction steps. This chemical silencing is lifted in the deprotection step, and the original reactivity is recovered. Numerous protecting groups are known for all possible functional groups, many of them with highly specific releasing mechanisms that enable orthogonal protecting and de-protecting [4,5]. One very attractive way to deprotect is via the action of light, either by direct absorption or by photocatalytic methods where energy or electron transfer propagates the electronic excitation to the protecting group. These groups are termed photoremovable (or photo releasable) protecting groups (PRPG) [6–10] that can also be designed as orthogonal PRPG [11,12]. Several publications have also chosen the term "photocages" [13–15]. The inverse effect, protecting groups that are specifically designed for reporting the deprotection process by light emission (chemiluminescent protecting groups, CLPG) has not yet been elaborated. Chemiluminescent compounds, in general, are small molecules that are activated by oxidation processes, mostly by the action of triplet oxygen to give highly unstable, strained

peroxides [16–19]. These peroxides are energetic molecules and are easily cleaved with the formation of electronically excited structures, mostly carbonyl or carbonyl-analogue nπ* states [20,21]. From the viewpoint of synthetic flexibility, only a few chemiluminescent systems are of possible relevance for developing CLPG applications because these systems must, beside CL emission, selectively release a predefined functional group simultaneously with light emission. We have now investigated the coumaranones as blue-emitting CL reagents and report their use as amino acid, pharmacophore and terpene PGs. These CL-active compounds were first published by Lofthouse [22]. In the initial paper, the authors described the luminescence for compound **1b** (Scheme 1) in dimethylformamide as a solvent in the presence of trimethylamine as "violet light clearly visible in daylight for ca. 30 min, and then fading slowly but still detectable in a darkened room after 70 h" [22].

**Scheme 1.** Structures of the parent benzofuran-2(3*H*)-one (**1a**), 5-chloro-coumaranone with a carbamate-linker (**1b**) published by Lofthouse et al. [22], and 5-fluoro-coumaranone with a urea-linker (**1c**) described by Krieg et al. [23].

Coumaranones offer a flexible basic structure that allows various options regarding incorporating biologically relevant compounds in the side-chains and variation of the aromatic scaffold. In recent years, the synthesis and CL-mechanism of 2-coumaranones have been thoroughly investigated [24–28]. In particular, the optimization of the synthetic protocol largely reduced the number of steps required. In general, a Tscherniak–Einhorn reaction is performed with an amide and glyoxylic acid monohydrate using trifluoroacetic acid or acetic acid containing 10% of sulfuric acid as solvent and catalyst. The subsequent addition of para-substituted phenol compounds generates the desired coumaranone in moderate to good yields.

The mechanism for the decomposition and CL of the substrates has been elucidated in the last decade [24–28]. The proposed steps start with deprotonation by a non-nucleophilic base (e.g., DBU), leading to the formation of a stabilized carbanion in the α-position to the lactone unit, which can react with molecular oxygen to form a 1,2-dioxetanone as a crucial intermediate. The subsequent elimination of $CO_2$ generates an electronically excited species, which emits a photon or transfers its electronic energy to a fluorescent emitter.

## 2. Materials and Methods

All reagents and solvents were purchased from commercial sources (*Alfa Aesar*, *Fischer Scientific*, *Carbolution*, *Arcos Organics* and *TCI*). The degree of purity of the compounds was >95% and they were used without any further treatment. [1]H- and [13]C-NMR spectra were recorded at 300, 500 or 600 MHz and 75, 125 or 150 MHz, respectively. The measurements were performed on a *Bruker* Avance II+ 600, *Bruker* Avance III 500, *Bruker* Avance III 499 and on a *Bruker* Avance II 300. The chemical shifts δ are reported in ppm downfield of the internal standard of TMS [δ ([1]H-NMR) = 0.00 ppm, δ ([13]C-NMR) = 0.00 ppm]. DMSO-d$_6$ [δ ([1]H-NMR) = 2.50 ppm, δ ([13]C-NMR) = 39.5 ppm] and CDCl$_3$ [δ ([1]H-NMR) = 7.24 ppm, δ ([13]C-NMR) = 77.2 ppm] were used as solvents. The coupling constant *J* is indicated in Hz. The fine structure is designated using the following abbreviations: s (singlet), d (doublet), t (triplet), q (quartet), quin (quintet), br (broad), ψ (pseudo) and m (multiplet). Infrared-spectra were measured with a Nicolet iS20 FTIR (Thermo Fischer Scientific, Waltham, MA). The wave numbers are categorized from 4000 to 800 cm$^{-1}$. The signals are listed with the following abbreviations: w (weak), m (medium), s (strong), vs. (very strong) and br (broad signal). Melting points of solid compounds were determined with a MP50 Melting Point

System (Mettler Toledo, Columbus, OH). High resolution mass spectra were measured with a MAT 900 S and with a LTQ Orbitrap XL (Thermo Fischer Scientific, Waltham, MA) via electron spray ionization (ESI). Flash chromatography was performed on silica gel 60 Å, particle size 0.035–0.070 mm (Macherey-Nagel, Düren, Germany). The chemiluminescence of **5A**, **5E**, **6** and **8** was measured with a FLS980 Photoluminescence Spectrometer (Edinburgh Instruments, Livingston, UK) with a photomultiplier tube-detector and a xenon lamp.

Synthesis of methyl ((5-chloro-2-oxo-2,3-dihydrobenzofuran-3-yl)carbamoyl)-(*S*)-phenylalaninate (**5A**): The synthetic protocol of Krieg et al. [23] was followed with small variations: (*S*)-3-phenyl-2-ureidopropanoic acid **2** (1.0 g, 4.80 mmol, 1.0 eq.) and glyoxylic acid monohydrate (0.44 g, 4.80 mmol, 1.0 eq.) were dissolved in trifluoroacetic acid (1.6 mL per 1 mmol urea compound). After 30 min, 4-chlorophenol (0.74 g, 5.76 mmol, 1.2 eq.) was added, and the solution was refluxed for 4 h. After cooling down to room temperature, the reaction was poured into a beaker with ice water (at least 4 times the volume of the reaction mixture) and the precipitated solid was filtered off, washed with water and dried overnight in vacuo over $P_2O_5$ in a desiccator. The raw product was purified by column chromatography on silica gel with ($CH_2Cl_2$/MeOH, 20:1 + 1% AcOH). Compound **5A** was obtained as a colorless powder with a yield of 1.32 g (3.52 mmol, 73%).

Synthesis of methyl ((5-chloro-2-oxo-2,3-dihydrobenzofuran-3-yl)carbamoyl)-(*S*)-phenylalaninate (**5E**): The synthetic protocol of Krieg et al. [23] was followed with small variations: (*S*)-methyl 3-phenyl-2-ureidopropanoate **3** (1.0 g, 4.50 mmol, 1.0 eq.) and glyoxylic acid monohydrate (0.41 g, 4.50 mmol, 1.0 eq.) were dissolved in trifluoroacetic acid (1.6 mL per 1 mmol urea compound). After 30 min, 4-chlorophenol (0.69 g, 5.40 mmol, 1.2 eq.) was added, and the solution was refluxed for 4 h. After cooling down to room temperature, the reaction was poured into a beaker with ice water (at least 4 times the volume of the reaction mixture) and the precipitated solid was filtered off, washed with water and dried overnight in vacuo over $P_2O_5$ in a desiccator. The raw product was purified by column chromatography on silica gel with ($CH_2Cl_2$/MeOH, 20:1). Compound **5E** was obtained as a colorless solid with a yield of 0.40 g (1.03 mmol, 23%).

Synthesis of 4-acetamidophenyl (5-chloro-2-oxo-2,3-dihydro-1-benzofuran-3-yl) carbamate (**6**): The synthetic protocol of Schramm [28] was followed with small variations: Glyoxylic acid monohydrate (0.52 g, 5.65 mmol, 1.01 eq.) and 4-acetamido-phenylcarbamate **4** (1.09 g, 5.62 mmol, 1.00 eq.) were dissolved in trifluoroacetic acid (1.6 mL per 1 mmol carbamate compound). After 5 min, 4-chlorophenol (0.86 g, 6.74 mmol, 1.20 eq.) was added and the solution was stirred at room temperature for 1 day. After that the reaction was poured into a beaker with ice water (at least 5 times the volume of the reaction mixture) and the precipitated solid was filtered off, washed with water and dried overnight in vacuo over $P_2O_5$ in a desiccator. The raw product was purified by column chromatography on silica gel with ($CH_2Cl_2$/MeOH 8:1). Compound **6** (0.26 g, 0.73 mmol, 13%) was obtained as a colorless solid.

Synthesis of (1*R*,2*S*,5*R*)-2-isopropyl-5-methylcyclohexyl (5-chloro-2-oxo-2.3-dihydrobenzofuran-3yl)carbamate (**8**): The synthetic protocol of Schramm [28] was followed with small variations: Glyoxylic acid monohydrate (0.46 g, 5.02 mmol, 1.0 eq.) and L-menthol carbamate **7** (1.00 g, 5.02 mmol, 1.0 eq.) were dissolved in trifluoroacetic acid (1.6 mL per 1 mmol carbamate compound). After 5 min, 4-chlorophenol (0.77 g, 6.02 mmol, 1.2 eq.) was added and the solution was stirred at room temperature for 3 days. After that the reaction was poured into a beaker with ice water (at least 5 times the volume of the reaction mixture) and the precipitated solid was filtered off, washed with water and dried overnight in vacuo over $P_2O_5$ in a desiccator. The raw product was purified by column chromatography on silica gel with (toluene/MeOH 100:1 + 1% AcOH). Compound **8** (0.45 g, 1.23 mmol, 25%) was obtained as a colorless solid.

## 3. Results

### 3.1. Syntheses of Coumaranones

The syntheses of the urea- and carbamate-containing precursors were performed according to the synthesis protocols of Chegaev et al. [29], Nicolas et al. [30] and Graf [31], respectively. The two urea compounds **2** and **3** were obtained in quantitative yields, the carbamates of paracetamol **4** and L-menthol **7** with yields of 72% and 88%, respectively. The coumaranones with a urea substructure **5A** and **5E** were synthesized according to the procedure by Krieg et al. [23] (Scheme 2). Instead of stirring the solutions at room temperature for one or several days, the process was accelerated by refluxing the reaction mixture for 4–8 h. Longer periods of refluxing led to a decrease in yield and resulted in the almost complete hydrolysis of the esters.

**Scheme 2.** Synthesis of coumaranones **5A**, **5E**, **6** and **8**.

In the case of coumaranones with a carbamate substructure, paracetamol and L-menthol were chosen as protected compounds as they represent natural/pharmacologically valuable substrates and the corresponding 5-fluoro coumaranone derivatives are already known [28]. The application of the synthetic protocol and the use of 4-chlorophenol instead of 4-fluorophenol led to different results, however. Using acetic acid/sulfuric acid (9:1) as well as formic acid resulted in low yields. By using trifluoroacetic acid and a reaction time of 72 h, the highest yields of coumaranone **6** and **8** were 13% and 25%, respectively.

### 3.2. NMR Investigations of the Cleavage of the Coumaranones

In order to trace the reaction process and the required time for complete cleavage of the potential CLPG, proton NMR measurements were performed after constant reaction time intervals. In general, 20 mg of each substrate was dissolved in 0.7 mL DMSO-d$_6$ and 2 equivalents of DBU were added. As soon as the addition was complete and the solution became homogenous, a proton NMR spectrum was measured. Between each measurement the solution was saturated with oxygen gas.

Comparing **5E** and **5A**, the first major change after the addition of DBU can be seen in the area from 8 to 10 ppm where the NH protons vanished (Figure 1). Additionally, a visible change within the area of the aromatic protons (6.5 to 7.5 ppm) can be observed over the whole time period. The most important change is the decrease of the signal at 5.25 ppm, which represents the α-acidic proton of the benzofuranone moiety. The signal was expected to show up as a doublet instead of a doublet of doublets. Coalescence experiments verified the existence of rotamers for both substrates. According to the literature, this proton is removed by DBU during the cleavage process, and, at the end of the reaction, the corresponding carbon atom is oxidized to a carbonyl group. While this signal fully disappeared after 30 min in the case of **5A**, the ester derivative **5E** was oxidized significantly slower. Due to the fact that both experiments still showed the presence of rotamers and no proton or carbon signals of phenylalanine could be detected, it was assumed that the amino acid was not released from the initial coumaranone. In order to determine the exact structure of the unknown compounds, which are formed under the basic environment, additional decomposition experiments were performed, where the products were extracted, washed multiple times with water in order to remove excessive DBU, and spectroscopically characterized (see Supplementary Information for details). Since the aqueous workup might also cause side reactions, the corresponding spectra of time-interval measurements were compared to that of the products obtained after extraction.

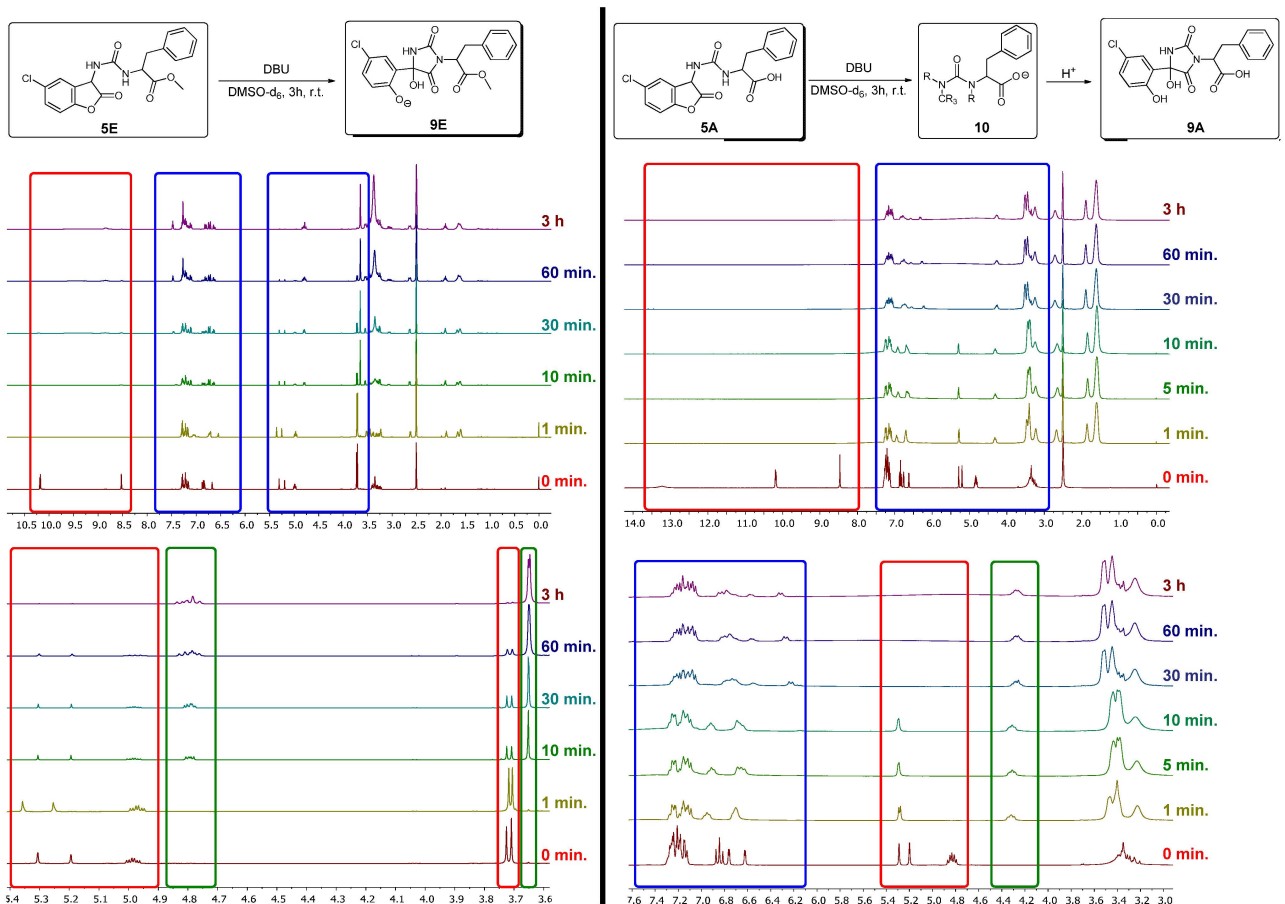

**Figure 1.** NMR measurements of the CL decomposition experiments with associated reaction equations. The image above shows the entire NMR spectra during the reaction; below is a section of a selected area where significant changes occur. (**left**); phenylalanine-coumaranone (**5E**) (**right**); phenylalanine-ester-coumaranone (**5A**). Blue boxes = change in signal intensity and occurrence of new signals; red boxes = reduction/disappearance of signals; green boxes = appearance of new signals.

After the decomposition experiment of **5E** was completed, the $^{13}$C-spectrum revealed that all nineteen carbon atoms were still present and most peaks split into two signals, indicating the presence of rotamers (see Supplementary Information). Also, the evaluation of the 2D-NMR spectra revealed that the α-lactone carbon became a quaternary carbon atom with a chemical shift of 81.8 ppm. This led to the assumption that, instead of a decomposition reaction (including the loss of $CO_2$), a rearrangement occurred. 2D-NMR and ESI experiments verified the hydantoin structure of **9E** as the main product of the experiment, proving that the α-carbon was oxidized. The urea undergoes a nucleophilic attack on the lactone carbonyl-carbon atom, leading to the formation of a five membered ring (Figure 1).

The 2D-NMR spectra of the decomposition experiment of **5A** led to different results. No products could be extracted from the basic solution. The $^{13}$C-NMR of the crude product (excluding the signals of DBU) showed that many NMR peaks were split into two signals showing the existence of rotamers (see Supplementary Information). Though a complete structure elucidation was not possible the evaluation of the 2D spectra led to fragment **10**. It is also noteworthy that no quaternary carbon could be found in the area of 80–100 ppm, indicating that, unlike in the case of **5E,** no hydantoin was formed as the main product. Only upon protonation with two equivalents of hydrochloric acid was it possible to extract a product (compound **9A**) whose structure could be fully resolved and represents the unprotected acid counterpart to **9E**. Therefore, it can be assumed that, before protonation, **5A** generated an open-chained carboxylate derivative as the main product.

While the urea-coumaranone derivatives did not release the corresponding amino acid, paracetamol **11** was successfully cleaved from coumaranone **6** accompanied by a strong, but very short, CL (Figure 2). Within a minute, the bright blue emission rapidly diminished and could no longer be seen with the human eye. Regarding the NMR-experiment, an increase in the aromatic signals of paracetamol **11** at 6.60 and 7.29 ppm appeared after the addition of DBU. The $CH_3$ group of the acetyl unit can be seen at 2.01 ppm and slightly shifts towards 1.97 ppm. Overall, the complete removal of the CLPG can be observed within a short period of time. Another important observation is the appearance of new signals in the aromatic region (excluding those of paracetamol) during cleavage. Although many new aromatic signals from several by-products appeared, no compounds other than **11** could be identified in the 2D NMR spectra after the decomposition experiment was completed.

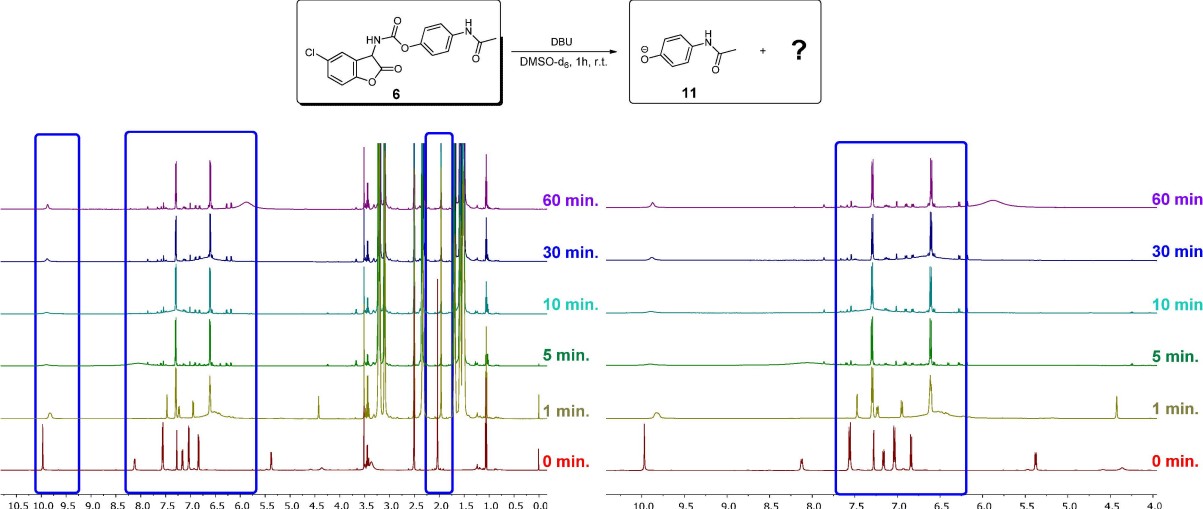

**Figure 2.** *Cont*.

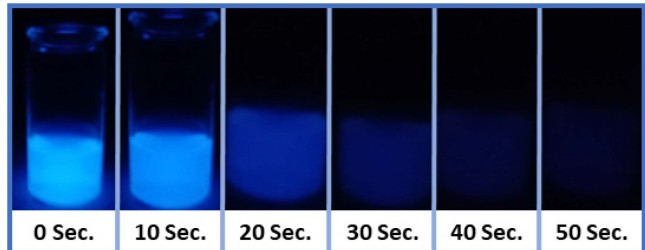

**Figure 2.** NMR measurements of the decomposition experiments with the paracetamol-coumaranone (**6**). (**top**, **left**); complete NMR-spectrum (**top**, **right**); enlarged extract of the aromatic area from 4.5–10 ppm. Blue boxes = change in signal intensity and occurrence of new signals (**bottom**); Decay of the CL of coumaranone **6**. 10 mg of **6** were dissolved in MeCN and a few drops of DBU were added. Within fifty seconds, the intensity of the initially strong and bright emission decreases and is no longer visible thereafter.

The results for L-menthol coumaranone **8** again differ with respect to the NMR experiment and CL properties. Although the decomposition process took longer compared to **6**, a bright and long-lasting CL was seen as soon as the sample was saturated with oxygen again in the NMR tube (Figure 3). After 20 min, the CL remained very faint and disappeared a few minutes later. The results of the NMR are in agreement with the intensity of the duration of the CL. While after one minute, the proton in the $\alpha$-position at 5.30 ppm and the NH proton at 8.25 ppm have already disappeared, there are still visible changes in the aromatic region (6.0–7.5 ppm) and in the range of 0.4–1.2 ppm. The signal of the CH group geminal to the oxygen atom of the menthol molecule (4.34 ppm) shows a slight shift to higher ppm values after completion of the oxidation process, which does not correspond to free L-menthol. 2D NMR allowed the elucidation of structure **12**, which confirms the successful decarboxylation and represents the emitting species of the CL process of coumaranones according to the literature [24–28]. In order to cleave the PG, the substrate was again decarboxylated with KOH in boiling ethanol, yielding L-menthol **14** as well as 5-chlorosalicylamide **13** (see Supplementary Information).

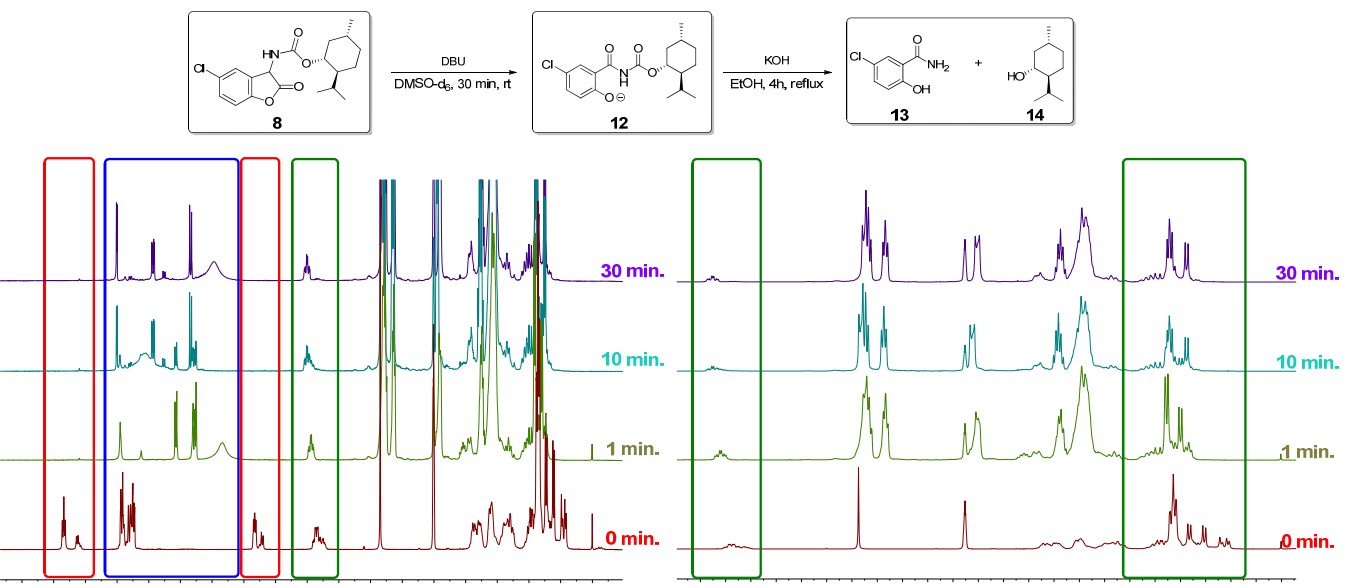

**Figure 3.** *Cont*.

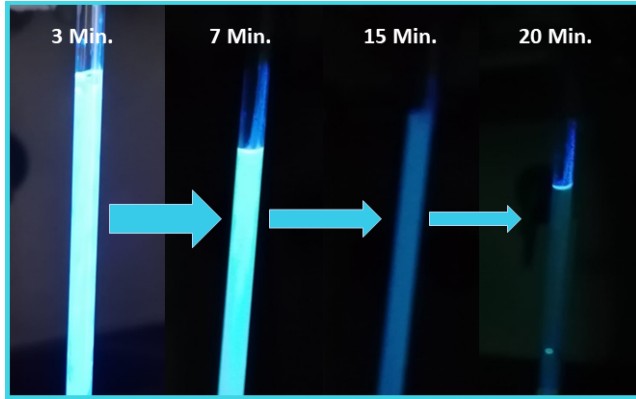

**Figure 3.** NMR measurements of the CL-decomposition experiment with the ʟ-menthol-coumaranone (**8**). (**top**, **left**); complete NMR-spectrum (**top**, **right**); enlarged extract of the area from 0.0–4.6 ppm. Blue boxes = change in signal intensity and occurrence of new signals; red boxes = reduction/disappearance of signals; green boxes = appearance/shift of new signals. (**bottom**); photos of the chemiluminescence of **8** during saturation with oxygen.

## 3.3. Chemiluminescence

Emission and excitation experiments were carried out to investigate the photophysical properties of the urea/carbamate coumaranones (**5A**, **5E**, **6** and **8**) and their reaction products. Each compound was measured with an initial substrate concentration of $c = 10^{-2}$ mol/L in acetonitrile or DMF. At the beginning of each experiment, 50 equivalents of DBU were added to 1 mL of the stock solution in a cuvette, which was then homogenized by shaking. The measurements were then performed instantly and monitored over several scans. The cuvette was never sealed during all measurements in order to prohibit increasing oxygen deficiency. Figure 4 shows the emission and excitation spectra of **5E**. The chemiluminescence has a maximum at 436 nm and diminishes very fast within the first two scans and completely fades after the twelfth measurement (Figure 4a). This is in accordance with the results in the literature [23] stating that the urea-coumaranones have a short and strong chemiluminescence in comparison to carbamate-coumaranone derivatives.

Performing multiple emission scans afterwards revealed that an absorbing species is produced that fluoresces with a maximum at 435 nm. Therefore, a second CL experiment, with an additional excitation wavelength of 370 nm, was performed (Figure 4b). Firstly, the diminishing chemiluminescence can be observed, while, after 50 scans, the photoluminescence increases over time. The emission is slightly red-shifted with an emission maximum at 436 nm. The third spectroscopic approach uses the excitation scan (Figure 4c, emission wavelength: 430 nm) in order to trace the changes in absorption of the previously observed fluorescent species. Again, within the 100 scans, an absorbing species with a maximum at 402 nm can be seen, which slowly decreases, and then a new species with a maximum at 396 nm appears.

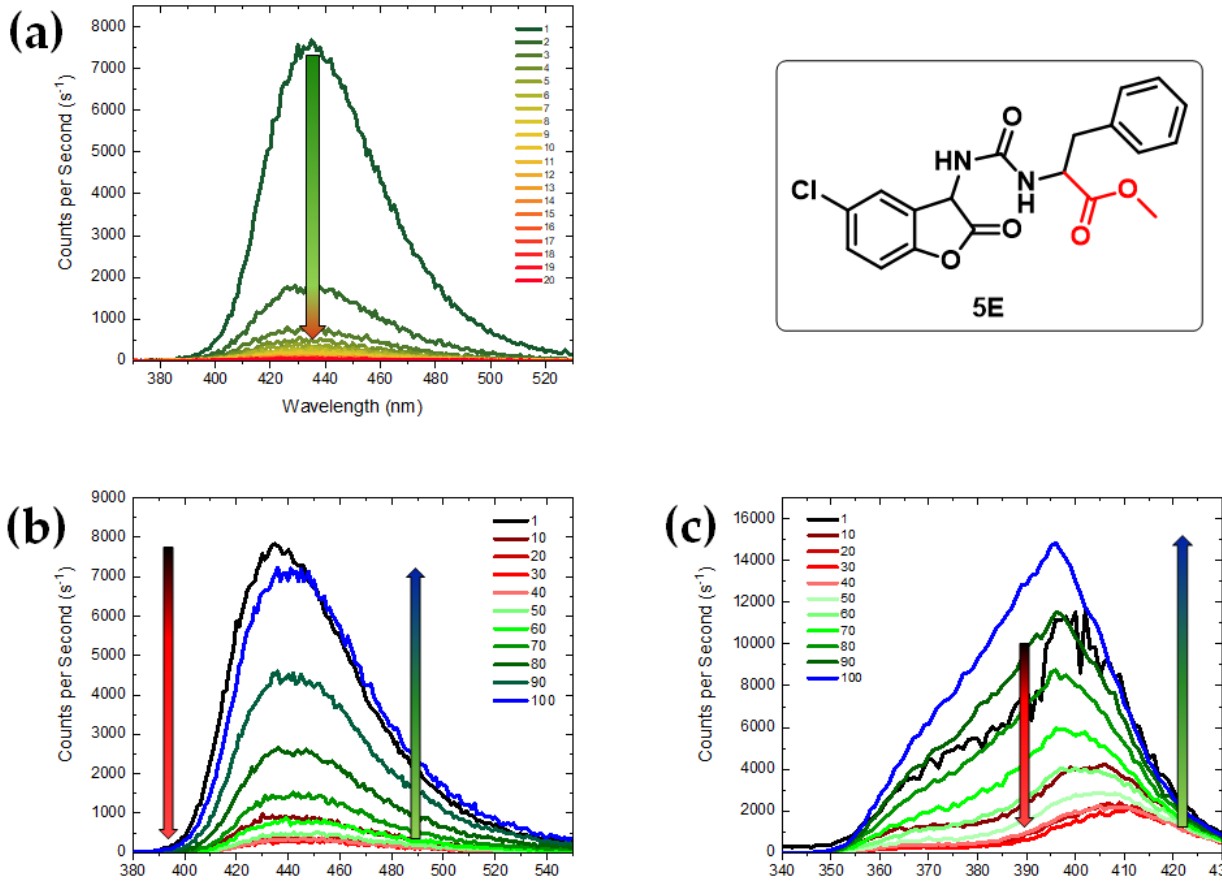

**Figure 4.** Emission and excitation spectra of **5E** in acetonitrile (every scan took 45 s): (**a**) emission scans of CL (i.e., without external excitation, Slit: 10), the arrow indicates the decrease in the CL; (**b**) emission scans during CL reaction (external excitation $\lambda_{Ex}$ = 370 nm, Slit: 1.0). The first scan was performed after addition of DBU. The left red arrow indicates the decrease in CL after 40 scans. Starting from the 50th scan, the right green arrow indicates the increase in photoluminescence; (**c**) excitation scans during CL reaction ($\lambda_{Em}$ = 430 nm, Slit: 1.0). The first scan was performed after addition of DBU. The left red arrow indicates the decrease in an absorbing species after 40 scans. Starting from the 50th scan, the right green arrow indicates the increase in absorption of a new species.

The same three spectroscopic experiments were performed for **5A** (Figure 5). Despite the fast decay of the chemiluminescence (5a), the emission and excitation measurements show a substantial contrast to those of **5E**. In both cases, the absorbing and emitting species first show an increase and, within the 100 scans, a minor decrease, until an equilibrium is reached. The emission maximum remains at 435/436 nm (5b), while the absorption maximum shifts from 381 nm to 376 nm (5c). These experiments suggest that the fluorescent species of **5A** is structurally related to that of **5E** but is formed within a few seconds and even overlaps its own CL (5b).

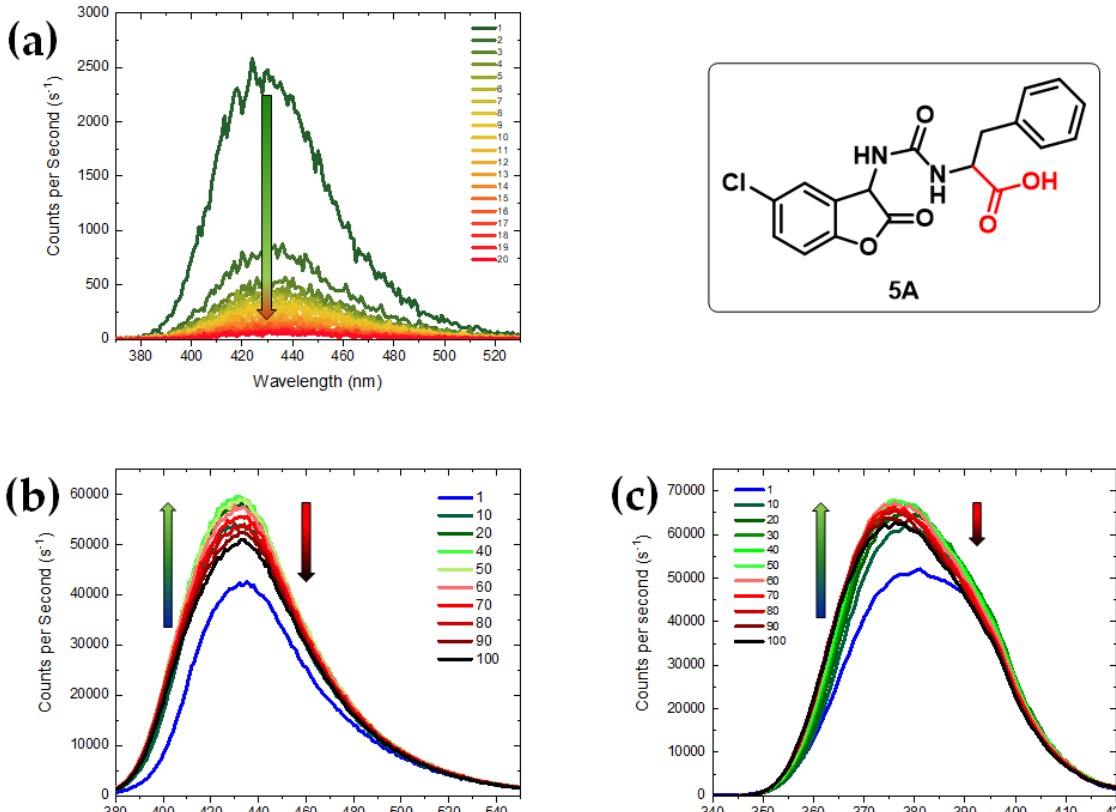

**Figure 5.** Emission and excitation spectra of **5A** in acetonitrile (every scan took 45 s): (**a**) emission scans of CL (i.e., without external excitation, Slit: 10), the arrow indicates the decrease in the CL; (**b**) emission scans during CL reaction (external excitation $\lambda_{Ex}$ = 370 nm, Slit: 1.0). The first scan was performed after addition of DBU. The left green arrow indicates the initial CL (scan 1) and the increase in photoluminescence after 40 scans. Starting from the 50th scan, the right red arrow indicates the small decrease in photoluminescence; (**c**) excitation scans during CL reaction ($\lambda_{Em}$ = 430 nm, Slit: 1.0). The first scan was performed after addition of DBU. The left green arrow indicates the increase in an absorbing species after 40 scans. Starting from the 50th, scan the right red arrow indicates a minor decrease in absorption.

In the case of **6**, the NMR experiments clearly showed the release of paracetamol, which is why different photophysical properties were expected. The chemiluminescence of **6** is clearly visible to the human eye and very intense, but vanishes within one minute, which is confirmed by the initial emission scans showing a maximum at 406 nm after one scan. Afterwards, no further emission can be detected (Figure 6a). Several emission scans with an excitation wavelength of 495 nm showed an increasing and strong photoluminescence with a maximum at 558 nm. However, while the CL decreased after one scan, the formation of the fluorescent species takes much longer and reaches an equilibrium only after 19 min (Figure 6b). In agreement with these results, the excitation scans (Figure 6c, $\lambda_{Em}$ = 555 nm) show a very strong absorption at 449 nm, which shifts slightly towards 452 nm after the first scan. During the decay of the first absorbing species, a new slight increase in absorbance is observed after the second scan with a maximum at 505 nm, creating an isosbestic point at 483 nm. Since several aromatic by-products could be seen in the NMR experiments of **6**, it remains unknown which one causes the observed absorption and fluorescence.

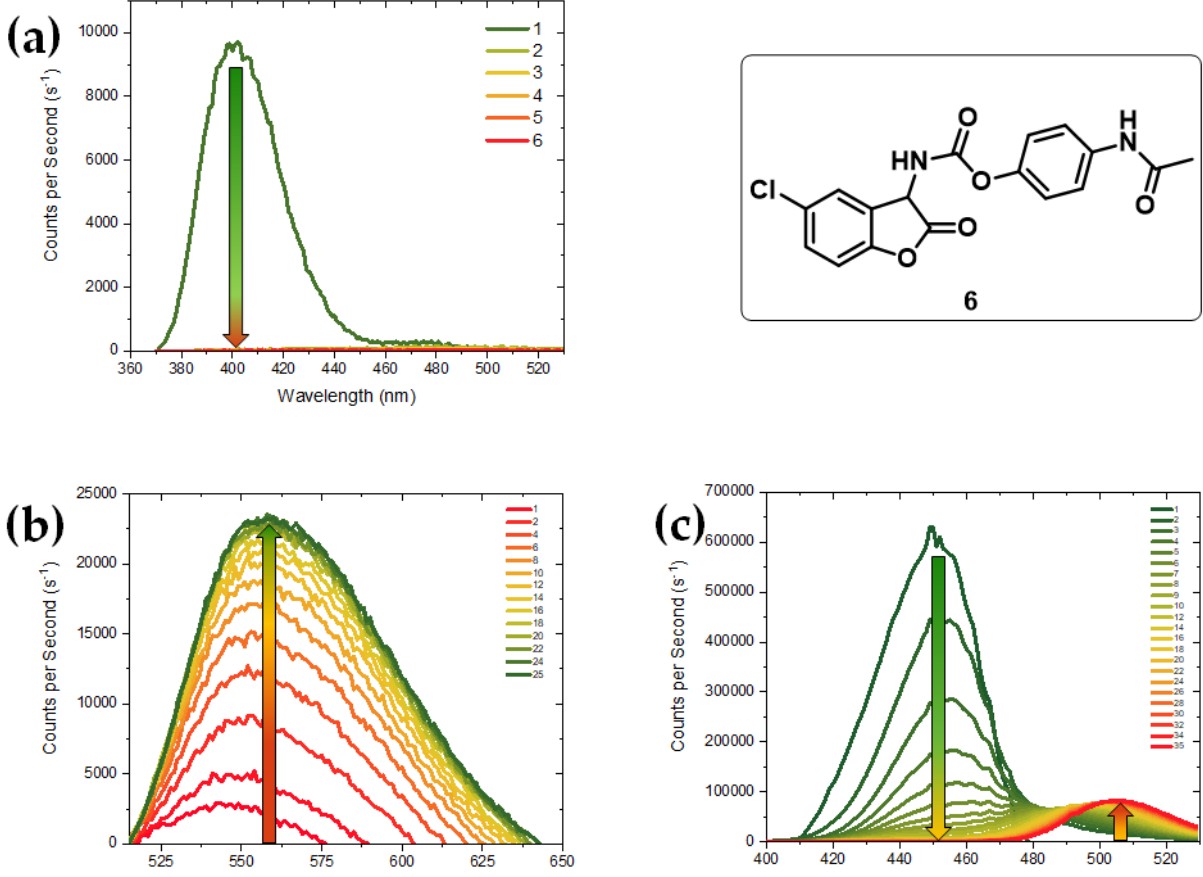

**Figure 6.** Emission and excitation spectra of **6** in DMF (every scan took 45 s): (**a**) emission scans of CL (i.e., without external excitation, Slit: 5.0), the arrow indicates the decrease in the CL; (**b**) emission scans during CL reaction (external excitation $\lambda_{Ex}$ = 495 nm, Slit: 1.5). The first scan was performed after addition of DBU. The arrow indicates the increase in a fluorescent species within 25 scans; (**c**) excitation scans during CL reaction ($\lambda_{Em}$ = 555 nm, Slit: 1.5). The first scan was performed after addition of DBU. The left green arrow indicates the decrease in an absorbing species after 20 scans. Starting from the 22nd scan, the right orange/red arrow indicates a minor increase in absorption at higher wavelengths.

The CL of **8** is initially very strong and bright and persists for about 21 min before fading (Figure 7a). After the 8th scan, the maximum shifts from 426 nm to 440 nm. Based on previous NMR experiments, it is known that compound **12** is formed during the CL reaction in an alkaline solution. Therefore, only one emission and excitation scan was performed to investigate the photophysical properties of this product. After completion of the CL reaction, a broad photoluminescence with a maximum at 463 nm could be detected (Figure 7b, $\lambda_{Ex}$ = 420 nm). An excitation scan with $\lambda_{Em}$ = 470 nm showed the corresponding absorption maximum at 415 nm (Figure 7c).

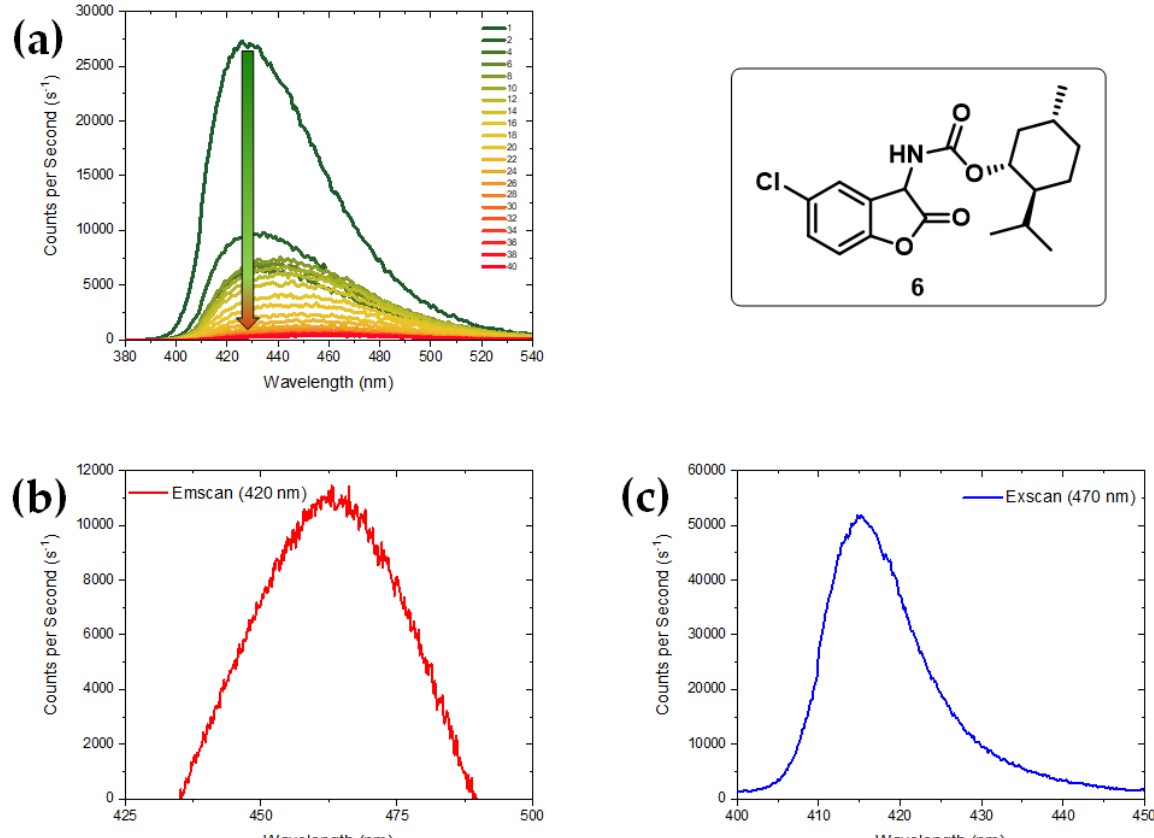

**Figure 7.** Emission and excitation spectra of **8** in acetonitrile (every scan took 45 s): (**a**) emission scans of CL (i.e., without external excitation, Slit: 0.1), the arrow indicates the decrease in the CL; (**b**) emission scan of **8** after CL was finished (external excitation $\lambda_{Ex}$ = 420 nm, Slit: 0.5); (**c**) excitation scan of **8** after CL was finished ($\lambda_{Em}$ = 470 nm, Slit: 1.0).

### 3.4. Mechanistical Considerations for **5A** and **5E**

Considering the previous results, the excitation experiments of **5A** and **5E** show that the fluorescent species appears to be structurally similar for both substrates, as the emission maxima differ only by 5–6 nm. Since the hydantoin structure of **9E** is the main product of **5E**, and the excitation experiments first show no and subsequently a steady growth in photoluminescence, it can be assumed that a slow equilibrium between **15-I** and an open chained oxoacetyl phenolate species (**15-IV**) takes place and shifts towards the latter after about 30 min (Scheme 3). Via tautomerization, the phenolate shifts the electron density towards the carbonyl oxygen of the urea moiety, which causes ring opening (**15-II**). Subsequent deprotonation (**15-III**) and rearomatization generates an oxoacetyl phenolate (**15-IV**), which is responsible for the observed fluorescence.

Compound **5A** shows an immediate and intense fluorescence within a few seconds. The only structural difference is the carboxylic acid, so this group has a major influence on the formation of the fluorescent species. Due to the basic environment, the carboxylate (**16-I**) is formed and initiates a nucleophilic attack and ring expansion. This leads to a shift of the electron density towards the carbonyl oxygen of the urea moiety (**16-II**). After the deprotonation of the alcohol (**16-III**) and the formation of the carboxylate, the oxoacetylphenolate analogue (**16-IV**) is formed. Since the carboxylate induces rapid ring expansion (five- to six-membered ring), the fluorescent species is formed much faster.

**Scheme 3.** Proposal for the formation of the fluorescent species from **5E** (**left**) and **5A** (**right**). The rate of the tautomerization processes differs due to the ester/free carboxylic acid.

## 4. Discussion

Reviewing coumaranones for the concept of CLPGs, two important observations were made: (a) the nucleofuge has a major influence on both the CL properties and subsequent cleavage. While paracetamol, which generates a phenolate upon cleavage, is rapidly released, alkyl alcohols and amines (in these cases, L-menthol and phenylalanine) are not released but remain as decarboxylated or oxidized products; (b) urea-coumaranones rearrange to stable hydantoins, which can also be isolated after work-up processes. If the carboxylic acid of phenylalanine has not been protected, the formation of an open-chain product is preferred, which can be easily converted into the corresponding hydantoin by neutralization of the alkaline solution.

The NMR experiments with the carbamate coumaranones **6** and **8** showed a fast oxidation process accompanied by bright CL, the duration of which correlated with the reaction rate. However, in the case of **8**, it was necessary to subsequently heat the benzoyl carbamate **12** in ethanol with KOH to release the substrate. Therefore, the CLPG of **8** can be regarded as a two-step PG, where only after completion of the CL reaction does the second decarboxylation of the carbamate yield the unprotected substrate. Therefore, this PG gives additional stability against strong non-nucleophilic bases and requires another reaction step in the overall synthesis.

The luminescence experiments show that the urea-coumaranones have a weak and short CL but develop a strong and potent photoluminescent species during the oxidation process, so that the designation FPG seems appropriate for these two examples. The formation of the oxoacetyl phenolate is clearly dependent on the carboxylic acid/ester of phenylalanine, as the alkaline environment allows the deprotonation of the free acid of **5A** and increases the reaction rate for the formation of the fluorescent compound. Taking the NMR experiments into account, **5E** is oxidized much slower than **5A**. The conversion of **5A** to the FPG **9A** is fast enough that the chemiluminescence is overlapped by the photoluminescence.

In view of these results and the detailed mechanistic investigations in the literature [24–28], the entire decomposition of 2-coumaranones needs to be revised with regard to the derivative studied (Scheme 4). Carbamate derivatives **6** and **8** both show a bright CL, the duration of which depends on how fast the release/oxidation of the substrate occurs. The nucle-

ofuges themselves also play an important role, as only paracetamol was released, while L-menthol remains bound to the decarboxylated benzoylcarbamate **12**, which required an additional reaction to completely cleave off the PG (Scheme 4, green box). Therefore, it can be concluded that the phenolate formed during decarboxylation requires stabilized leaving groups, making it a selective nucleophilic group for the concept of CLPGs.

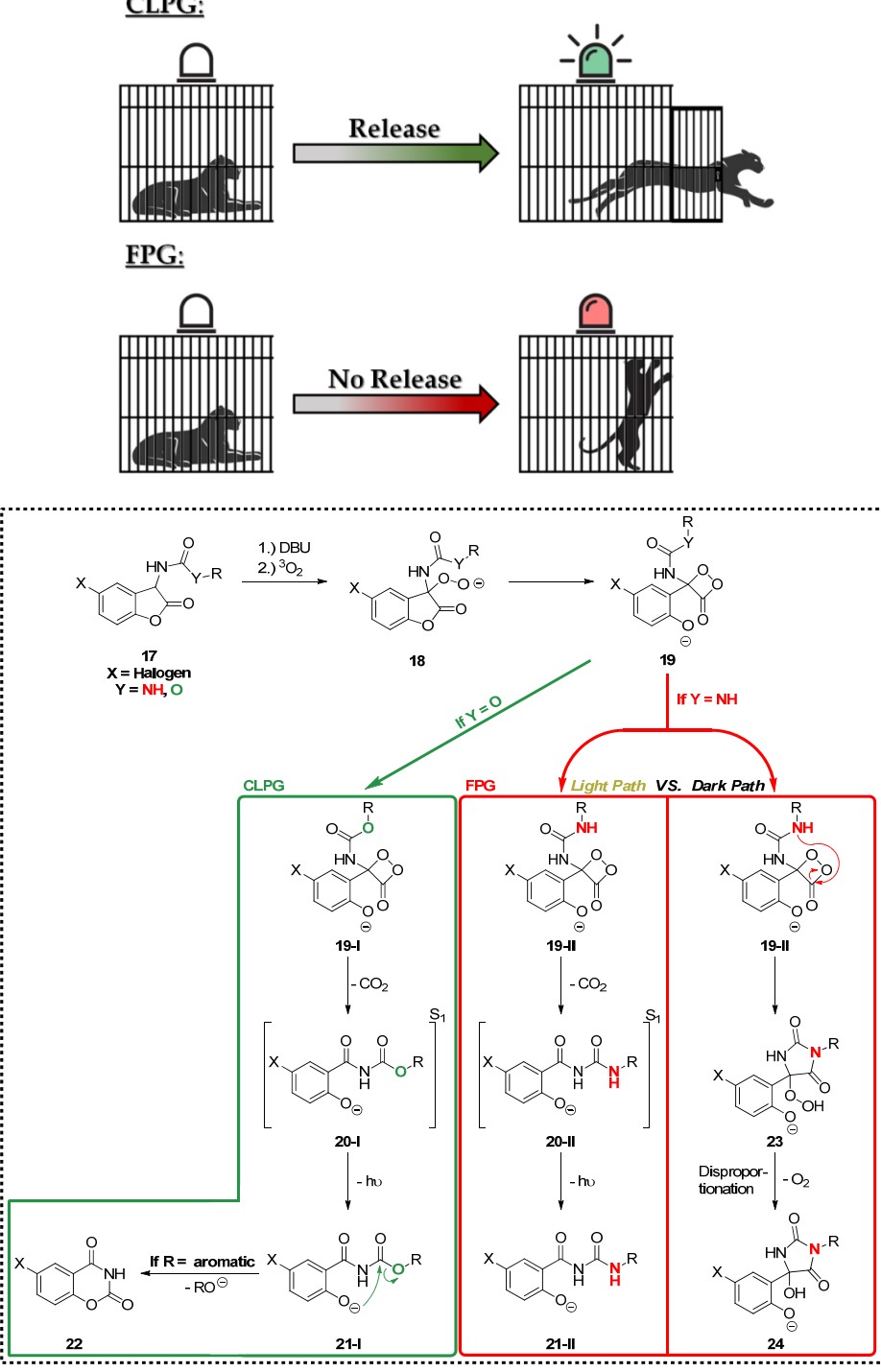

**Scheme 4.** (**Top**) Illustration of the concept of a **CLPG** and **FPG**, respectively, with the CL-photon emitter (the lamp), a protecting group (the cage) and a protected compound (sleeping panther). In the case of a **CLPG** (**top**), the release of the protected molecule is accompanied by light emission. In the case of the **FPG** (**bottom**), the molecule remains connected to the protecting group and shows new fluorescence properties. (**Bottom**) Mechanism comparison between urea and carbamate coumaranones.

When compared to the carbamate derivatives, the urea derivatives **5A** and **5E** show a weaker and short-lived CL (Scheme 4, red box, light path). The preferred route is a nucleophilic attack of the urea moiety (most likely after the formation of the 1,2-dioxetanone **19-II**), which prohibits the loss of $CO_2$ and the required energy for a CL to occur (Scheme 4, red box, dark path). Instead, a 5-hydroperoxy hydantoin **23** is generated, which eventually leads to the corresponding 5-hydroxy hydantoin **24**. Thus, **5A** and **5E** can be considered FPGs instead of CLPGs due to the assumed reaction mechanism and the obtained products, which show a bright and visible photoluminescence.

## 5. Conclusions

Two new urea-coumaranones with a proteinogenic amino acid (**5A** and **5E**) and the 5-chloro coumaranone derivatives of paracetamol **6** and L-menthol **8** were studied. The results of the NMR and luminescence studies demonstrate that the CLPG concept is strongly dependent on the properties of the nucleofuge and the side-chain in general. While the phenolate of paracetamol is rapidly released in the CL reaction of **6** with DBU, L-menthol shows a strong CL as well but is not cleaved from the PG unless an additional decarboxylation with KOH is carried out. Phenylalanine is not released from the urea-coumaranones due to the low leaving group potential. Instead, a nucleophilic attack of the urea on the critical 1,2-dioxetanone intermediate leads to the formation of hydantoin **9E** (starting from **5E**) or an open-chained carboxylate species **10** (starting from **5A**). Compound **10** is converted to the corresponding hydantoin **9A** when treated with an acid. According to the literature, the imide-protected amine within the hydantoin PG can be cleaved either enzymatically or chemically with LiOH or KOH [32]. Spectroscopic experiments showed that the CL of both **5A** and **5E** decays rapidly and is rather weak. After CL, photoluminescence could be detected. While the photoluminescence of **5A** was seen immediately after addition of DBU, the initial formation of the fluorescent species of **5E** took 38 min. Thus, the protection of the carboxylic acid has a major influence on the formation of the fluorescent species. Given the only slight differences in the absorption and emission properties of the photoluminescence of **5A** and **5E**, it is assumed that, in both cases, the structure of the fluorophore is an open-chain oxoacetyl phenolate. Thus, we show that biologically or pharmaceutically relevant molecules from the terpene and amino acid families can be incorporated into urea-/carbamate-linked coumaranones. These compounds, depending on their specific structures and especially the leaving group abilities, serve as CLPG or as potential biomarkers by fluorescence switching (FPG).

**Supplementary Materials:** The following supporting information can be downloaded at: https://www.mdpi.com/article/10.3390/photochem3030023/s1, Figure S1: NMR- and IR-spectra of **5A**; Table S1: 1D and 2D-NMR data of **5A** in DMSO-$d_6$; Figure S2: NMR- and IR-spectra of **5E**; Table S2: 1D and 2D-NMR data of **5E** in DMSO-$d_6$; Figure S3: NMR- and IR-spectra of **6**; Table S3: 1D and 2D-NMR data of **6** in DMSO-$d_6$; Figure S4: NMR- and IR-spectra of **8**; Table S4: 1D and 2D-NMR data of **8** in CDCl$_3$; Figure S5: NMR- and ESI-spectra of **9E**; Table S5: 1D and 2D-NMR data of **9E** in DMSO-$d_6$; Figure S6: NMR-spectra of **9A**; Table S6: 1D and 2D-NMR data of **9A** in DMSO-$d_6$; Figure S7: NMR-spectra of **5A-DE** before protonation (Contaminated with DBU); Figure S8: NMR-spectra of **12** (Contaminated with DBU); Table S7: 1D and 2D-NMR data of **12** in DMSO-$d_6$.

**Author Contributions:** Conceptualization, T.L., A.G.G. and M.S.W.; writing original draft preparation, T.L. and A.G.G.; research, T.L. and R.H.; supervision, A.G.G.; methodology, L.S. and N.T.F. All authors have read and agreed to the published version of the manuscript.

**Funding:** This research received no external funding.

**Data Availability Statement:** Data are contained within the article or Supplementary Material.

**Acknowledgments:** We thank Mathias Wickleder, Laura Straub, and Niko T. Flosbach for the luminescence measurements of the samples.

**Conflicts of Interest:** The authors declare no conflict of interest.

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
