# Peer review of "5-Chlorocoumaranone-Conjugates as Chemiluminescent Protecting Groups (CLPG) and Precursors to Fluorescent Protecting Groups (FPG)"

_2673-7256, doi:10.3390/photochem3030023_

Round 1

Reviewer 1 Report

In this work, Lippold and co-workers studied the potential of 5-5-Chlorocoumaranone-Conjugates as CL protecting groups (CLPG). More specifically, the authors studied both urea- and carbamate-coumaranones, and tested the principle for the release of phenylalanine, paracetamol and l-menthol. While paracetamol was successfully released as a direct consequence from the CL reaction, that was not the case for neither l-menthol and phenyalanine. The objetive and topic of this work is a good fit for this journal and its readership. However, there are still relevant issues that must be addressed in Major Revision:

-In Introduction section, the authors claim that deprotecting process by CL (CLPG) has not been yet elaborated. How does the present work differs, in terms of CL emission after deprotection, from the work of other authors such as Shabat et al (for ex., DOIs:https://doi.org/10.1039/D3SC02386A; https://doi.org/10.1002/chem.202300422)?

-In the introduction section, the authors state that only some systems are of possible relevance for developing broad CLPG applications. What systems are those and/or what specific characteristics do they need to possess to be of potential use?

-The CL reaction mechanism of coumaranone should be presented in the Introduction, in a schematic form;

-The authors should discuss what biologically-related applications can be compatible with the triggering requirements of this type of CL substrate (addition of non-nucleophilic base);

-For each CL substrate, the authors should present a figure in which are overlapped the normalized CL spectrum and fluorescent spectrum of the spent reaction;

-It is not clear what sub-figures b and c, of Figures 4 to, are actually measuring. Is it the fluorescence and excitation spectra during the CL reaction, of the substrate without base, or after the CL reaction?

-If the authors want to scan changes in the absorption, why did they not just measure the absorption spectra of these substrates by UV-Vis spectroscopy, instead of measuring excitation spectra? The latter approach is less informative than the former;

-Why the relevant noise in emission and excitation spectra?

-Can the conditions used to cleave 12 into 13 and 14, also be used to cleave 9E and 9A?

- I am confused by the use of the coumaranone CL system as CLPG. When considering available literature (for example, DOI: 10.1039/C3PP50345C), there is no actual expectancy that this CL reaction leads to the release of specific moieties of the initial substrate. In fact, the surprising result appears to be more that 6 did lead to release of the protected substrate, than neither 5A, 5E nor 8 leading to release of a substrate as a direct result of a CL reaction;

-Following on the previous point, more mechanistic focus should be given to 6, in order to explain why this substrate is such an outlier (specially compared to 8);

-The title, abstract and conclusion appear to be somewhat misleading as they seem to indicate that using 5-Chlorocoumaranone-Conjugates, in general, as CLPG was shown to be a viable concept. However, from the 4 studied conjugates, 3 did not lead to direct release of protected substrate from a CL reaction. Just one CL substrate did lead to that release but, at the moment, it is not clear if this result is not specific to paracetamol. So, the general viability of this concept, involving this type of molecules, was not demonstrated. This must be very clear in the above-mentioned sections.

-What are the slits used in the CL and photoluminescent measurements?

DOI

Author Response

Dear Editors of Photochem,

we herewith submit a revision of our submitted manuscript entitled “5-Chlorocoumaranone-Conjugates as Chemiluminescent Protecting Groups (CLPG) and Time-Delayed Releasing Groups: Urea- Vs. Carbamate Coumaranones with the manuscript number photochem-2505310.

We would like to thank all three referees for their fair and constructive comments and suggestions. The changes / additions made by us are explained and highlighted in the letter as well in the revised manuscript in yellow color.

Reviewer 1 (reviewer comments and suggestions in italics):

In this work, Lippold and co-workers studied the potential of 5-5-Chlorocoumaranone-Conjugates as CL protecting groups (CLPG). More specifically, the authors studied both urea- and carbamate-coumaranones, and tested the principle for the release of phenylalanine, paracetamol and l-menthol. While paracetamol was successfully released as a direct consequence from the CL reaction, that was not the case for neither l-menthol and phenyalanine. The objetive and topic of this work is a good fit for this journal and its readership. However, there are still relevant issues that must be addressed in Major Revision:

We thank the reviewer for the positive comments on the fit of our manuscript to this journal and the comments.

-In Introduction section, the authors claim that deprotecting process by CL (CLPG) has not been yet elaborated. How does the present work differs, in terms of CL emission after deprotection, from the work of other authors such as Shabat et al (for ex., DOIs:https://doi.org/10.1039/D3SC02386A; https://doi.org/10.1002/chem.202300422)?

This comment by the reviewer is absolutely accurate and we are sorry that we seem to have given the wrong impression here. We do not want to create the impression that no protecting groups exists that are cleaved with concomitant CL or vice versa no CL systems exist that lead to the release of potentially interesting leaving groups. Beside the paper mentioned by the reviewer (that is linked to a – well-studied and classical - enzymatic activity visualization by CL) there are other reports in the literature, also by the Schramm group that has intensively studied this CL-system. What we would like to do here is to focus on a PG that is able to report the release spatiotemporal by light emission without external excitation, i.e. a CLPG which might by a useful concept for all kind of PG where the release kinetic is measurable by photon counting.

To avoid giving the wrong impression, we have modified the sentence in line 16 in the Abstract to:
“This principle is proposed and investigated for phenylalanine…………….”

-In the introduction section, the authors state that only some systems are of possible relevance for developing broad CLPG applications. What systems are those and/or what specific characteristics do they need to possess to be of potential use?

What we wanted to say is that there are relatively few chemiluminescent systems that can be easily coupled to interesting functional groups that are released during the CL process. Dioxetanes, for example, are great CL compounds that can hardly be converted to a broad set of CLPG´s, which is also true for several other CL families. The coumaranones belong to a very flexible class of CL precursors that are easily combined with alcohols, amines, phenols etc. In this sense, there are only few classes of CL structures that are in principle feasible as CLPG. To avoid giving the wrong impression, we have modified the sentence in line 45 in the Introduction to: “……..for developing CLPG applications, because these systems must, beside CL emission, selectively release a pre-defined functional group simultaneously with light emission”

-The CL reaction mechanism of coumaranone should be presented in the Introduction, in a schematic form;

What we have decided with respect to the literature mechanism of coumaranone oxidative decay (leading to CL and FG release), was to integrate this into Figure 4 (the green box). We still think that this is the best place to present, discuss and compare the decay / CL mechanism with the results for our compounds. We have added to line 395: “In view of these results and the detailed mechanistic investigations in the literature [23-26], the entire decomposition mechanism…………….”

-The authors should discuss what biologically-related applications can be compatible with the triggering requirements of this type of CL substrate (addition of non-nucleophilic base);

We have added a final sentence to the very end of the conclusion relate to the release of biologically relevant molecules by the CLPG approach: “Thus, we could show that biologically or pharmaceutically relevant molecules from the terpene and amino acid families can be incorporated into urea / carbamate-linked coumaranones. These compounds, depending on their specific structures and especially the leaving group abilities, serve as CLPG or as potential biomarkers by fluorescence switching (FPG).”

-For each CL substrate, the authors should present a figure in which are overlapped the normalized CL spectrum and fluorescent spectrum of the spent reaction;

The point is very well taken by the reviewer: overlapped CL und F spectra are useful to show if CL and F originate from the same species - or the F spectra originate from another species than the CL-active component. We think that we did clearly show this already from the spectra comparison a) versus b), especially in Figures 4,6,7, where the red-shift is depicted from the initial CL in a) to the terminal photo-luminescence spectra in b).

-It is not clear what sub-figures b and c, of Figures 4 to, are actually measuring. Is it the fluorescence and excitation spectra during the CL reaction, of the substrate without base, or after the CL reaction?

Both Figures b) and c) show the changes during the CL process, i.e. Figure b) shows an overlap between CL and photoluminescence (and thus the CL-F shift) and Figures c) show the excitation spectra buildup during the decomposition process. We have added also more info to the legends to make this clear.

-If the authors want to scan changes in the absorption, why did they not just measure the absorption spectra of these substrates by UV-Vis spectroscopy, instead of measuring excitation spectra? The latter approach is less informative than the former;

We have not applied this spectroscopical approach because we obtain to run all spectral information from the very same instrument. Of course, the reviewer is right in his advice but the information that we present is also complete for characterization of CL and F / excitation information. We are not able currently to run and add these desired spectra.

-Why the relevant noise in emission and excitation spectra?

The noise in the initial CL and photoluminescence could not be completely suppressed due to diffusion processes (despite homogenization by shaking the cuvette). Adjustments of the measurement parameters as well as autocorrection did not strongly optimize the quality of the spectra.

-Can the conditions used to cleave 12 into 13 and 14, also be used to cleave 9E and 9A?

Yes, as already described in the literature for the base-induced hydrolysis of hydantoins with substituents at the the N-imide nitrogen, the release of the PG is also possible also by action with aqueous KOH. We added the sentence to line 426: According to the literature, the imide-protected amine within the hydantoin PG can be cleaved either enzymatically or chemically with LiOH or KOH [32].
and the new reference [32]: Konnert, L., Lamaty, F., Martinez, J., Colacino, E. Recent Advances in the Synthesis of Hydantoins: The State of the Art of a Valuable Scaffold. Chem. Rev. 2017, 117, 13757-13809.

- I am confused by the use of the coumaranone CL system as CLPG. When considering available literature (for example, DOI: 10.1039/C3PP50345C), there is no actual expectancy that this CL reaction leads to the release of specific moieties of the initial substrate. In fact, the surprising result appears to be more that 6 did lead to release of the protected substrate, than neither 5A, 5E nor 8 leading to release of a substrate as a direct result of a CL reaction;

This comment by the reviewer is correct and we discuss the transition from a CL-system that, after light emission, changes into a FPG with the functional group still part of the molecule to a CL-system, where CL is connected to functional group release (but in no cases simultaneously). Because light emission and FG release is always temporally delayed, ALL compounds that we report here are in principle CLPG under right cleavage conditions.

-Following on the previous point, more mechanistic focus should be given to 6, in order to explain why this substrate is such an outlier (specially compared to 8);

This effect is due to a combination of high CL efficiency combined with good leaving group (phenolate) properties. We tried to indicate this in the green (as a “positive” color) box in Scheme 4.

-The title, abstract and conclusion appear to be somewhat misleading as they seem to indicate that using 5-Chlorocoumaranone-Conjugates, in general, as CLPG was shown to be a viable concept. However, from the 4 studied conjugates, 3 did not lead to direct release of protected substrate from a CL reaction. Just one CL substrate did lead to that release but, at the moment, it is not clear if this result is not specific to paracetamol. So, the general viability of this concept, involving this type of molecules, was not demonstrated. This must be very clear in the above-mentioned sections.

We agree with these comments and we have changed several information in the text accordingly:
The title was changed from:
 “5-Chlorocoumaranone-Conjugates as Chemiluminescent Protecting Groups (CLPG) and Time-Delayed Releasing Groups: Urea- Vs. Carbamate Coumaranones” to
“5-Chlorocoumaranone-Conjugates as Chemiluminescent Protecting Groups (CLPG) and precursors to Fluorescent Protecting Groups (FPG)”
In the Abstract, we have modified the sentence in line 16 in the Abstract to: “This principle is proposed and investigated for phenylalanine…………….”
In the Conclusion part, we have changed / added at the end the sentence: “Thus, we could show that biologically or pharmaceutically relevant molecules from the terpene and amino acid families can be incorporated into urea / carbamate-linked coumaranones. These compounds, depending on their specific structures and especially the leaving group abilities, serve as CLPG or as potential biomarkers by fluorescence switching (FPG).”

-What are the slits used in the CL and photoluminescent measurements?

The information of slit size was removed from the legends to all spectroscopic figures initially and we have now added again these data to the Figure legends.

Reviewer 2 Report

Some other halogen substitutions may be conducted. 

The DBU equivalent dependent manner must be checked.

It is quite confuse to understand explanatory notes in fig 4~7.

Moderate editing of English language required

Author Response

Dear Editors of Photochem,

we herewith submit a revision of our submitted manuscript entitled “5-Chlorocoumaranone-Conjugates as Chemiluminescent Protecting Groups (CLPG) and Time-Delayed Releasing Groups: Urea- Vs. Carbamate Coumaranones with the manuscript number photochem-2505310.

We would like to thank all three referees for their fair and constructive comments and suggestions. The changes / additions made by us are explained and highlighted in the letter as well in the revised manuscript in yellow color.

Reviewer 2 (reviewer comments and suggestions in italics):

We thank the reviewer for the positive comments on our manuscript.

Some other halogen substitutions may be conducted. 

There are already studies in the literature on variations of halide substitution at the aromatic core of the protecting group and chlorine is reported as the optimal substituent and synthetically easily accessible. We have therefor not tried to use other substituents in this study.

The DBU equivalent dependent manner must be checked.

This is a good advice that we will try to follow in future work. We have followed a literature protocol with high excess (50 eq.) of base in quantitative and 2 eq. in spectroscopic experiments (NMR studies).

It is quite confuse to understand explanatory notes in fig 4~7.

We have tried to make these legends easier to understand. It is, because of the complexity of the comparison between CL, photoluminescence and absorption, advisable to be more detailed in these descriptions.

Reviewer 3 Report

The manuscript is devoted to study of cleavage of phenylalanine, paracetamol and L-menthol protected with urea- and carbamate-coumarinones. The work is valuable for development of photoremovable protecting groups approach. Advanced experiments on NMR measurements of chemiluminescent decomposition experiments were performed, the formation of the fluorescent species was explained. The patterns of the decay path of protected compounds from the nature of the leaving group were revealed.

The manuscript can be accepted after minor revisions. On my opinion, only some improvements in Experimental part and Scheme 2 should be performed.

Lines 105 and 117. Melting points in comparison with published [27] data

Lines 129 and 141. Melting points in comparison with published [28] data

Line 143 A paragraph devoted to the experiments with DBU, KOH and so on should be given.

Scheme 2. Structure of glyoxylic acid monohydrate above the arrow. R1 and R2 are absent in the formulas, so writing R = H (2), R = Me (3); R = H (5A), R = Me (5B) would be preferable. Conversion of 4 into 6 takes 1 day according to Experimental part (line 124).

Lines 198 and 208 - 13C NMR spectrum.

Author Response

Dear Editors of Photochem,

we herewith submit a revision of our submitted manuscript entitled “5-Chlorocoumaranone-Conjugates as Chemiluminescent Protecting Groups (CLPG) and Time-Delayed Releasing Groups: Urea- Vs. Carbamate Coumaranones with the manuscript number photochem-2505310.

We would like to thank all three referees for their fair and constructive comments and suggestions. The changes / additions made by us are explained and highlighted in the letter as well in the revised manuscript in yellow color.

Reviewer 3 (reviewer comments and suggestions in italics):

The manuscript is devoted to study of cleavage of phenylalanine, paracetamol and L-menthol protected with urea- and carbamate-coumarinones. The work is valuable for development of photoremovable protecting groups approach. Advanced experiments on NMR measurements of chemiluminescent decomposition experiments were performed, the formation of the fluorescent species was explained. The patterns of the decay path of protected compounds from the nature of the leaving group were revealed.

We thank the reviewer for the positive comments on our manuscript.

The manuscript can be accepted after minor revisions. On my opinion, only some improvements in Experimental part and Scheme 2 should be performed.

We thank the reviewer for these critical comments, which we have taken into account in the following.

Lines 105 and 117. Melting points in comparison with published [27] data
Lines 129 and 141. Melting points in comparison with published [28] data

The melting points for all products 5A, 5E, 6, and 8 were added to the SI and to the main text. All these compounds are literature-unknown; the literature values for the 6 and 8 correspond to the fluorine-containing analogs.

Line 143 A paragraph devoted to the experiments with DBU, KOH and so on should be given.

The release part of the terpene 14 from the CLPG 8 was added to the SI because it is not relevant for the general message of the paper.

Scheme 2. Structure of glyoxylic acid monohydrate above the arrow. R1 and R2 are absent in the formulas, so writing R = H (2), R = Me (3); R = H (5A), R = Me (5B) would be preferable. Conversion of 4 into 6 takes 1 day according to Experimental part (line 124).

We totally agree with the comments by the reviewer. The use of R1 and R2 is misleading and false and we have deleted these numbers from the Scheme 2. The reaction time was also corrected to the correct value from the literature reference.

Lines 198 and 208 - 13C NMR spectrum.

We have added Figures of the corresponding 13C-NMR spectra now in the Supplementary Information (SI).

Round 2

Reviewer 1 Report

The manuscript could be accepted.

Reviewer 2 Report

Revised manuscript will meet criteria to publish the journal.